# Customized Imperialist Competitive Algorithm Methodology to Optimize Robust Miller CMOS OTAs

Egon Henrique Salerno Galembeck [1], Salvador Pinillos Gimenez [1,2] and Rodrigo Alves de Lima Moreto [1,*]

1  MTG2i Solutions, São Bernardo do Campo 09850-901, Brazil
2  Department of Electrical Engineering, FEI University Center, São Bernardo do Campo 09850-901, Brazil
*  Correspondence: rmoreto@mtg2isolutions.com

**Abstract:** The design and optimization of the analog complementary metal-oxide-semiconductor (CMOS) integrated circuits (ICs) are intrinsically complicated and depend heavily on the designer's experience, and are associated with very long design and optimization-cycle times. In addition, in order to the analog and radiofrequency (RF) CMOS IC work suitably in practice, it is necessary to perform robustness analyses (RAs) through Simulation Program with Integrated Circuit Emphasis (SPICE) simulations, which result in still-higher design and optimization cycle times and therefore represent the biggest bottleneck to the launching of new electronic products. In this context, this manuscript aims to present, for the first time, the use of a custom imperialist competitive algorithm (ICA) in order to reduce the design and optimization-cycle times of analog CMOS ICs. In this study, we implement some Miller CMOS operational transconductance amplifiers (OTAs) using the computational tool named iMTGSPICE, considering two different bulk CMOS IC manufacturing processes from Taiwan Semiconductor Company (TSMC) (180 nm and 65 nm nodes) and two evolutionary optimization methodologies of artificial intelligence, i.e., ICA and a genetic algorithm (GA). The main result obtained by this work shows that, by using an ICA-customized evolutionary algorithm to perform the design and optimization processes of Miller CMOS OTAs, it is possible to reduce the design and optimization-cycle times by up to 83% in relation to those implemented with the GA-customized evolutionary algorithm, achieving practically the same electrical performance.

**Keywords:** imperialist competitive algorithm; genetic algorithm; analog CMOS IC design; Miller CMOS OTA; robustness analyses

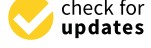



## 1. Introduction

In order to develop, design, and optimize analog and radiofrequency (RF) complementary metal-oxide-semiconductor (CMOS) integrated circuits (ICs), it is necessary to meet several design specifications and figures of merit (FoMs), such as the open-loop voltage gain ($A_{V0}$), unit voltage-gain frequency ($f_T$), phase margin (*PM*), slew rate (*SR*), etc., all at the same time. When these tasks are performed manually, they become very complex, boring, and difficult, and consequently demand long design and optimization-cycle times (DOCT) [1,2]. Besides, these activities are strongly dependent on the knowledge and experience of the designers [3–5]. Furthermore, these processes are extremely influenced by the aggressive and continuous effects of the variations of CMOS IC manufacturing processes and dimension reductions of metal-oxide-semiconductor field-effect transistors (MOSFETs), resulting in the appearance of several undesirable effects that are capable of degrading the electrical performance of MOSFETs and, consequently, the analog and RF CMOS ICs, such as the short-channel effect (SCE), thin-gate oxide-thickness fluctuations, etc., and subsequently generating undesirable variations in the mobility of the mobile charge carriers in the channel region of MOSFETs ($\mu_0$), threshold voltage ($V_{TH}$), increase in the leakage drain current ($I_{LEAK}$), subthreshold slope (*SS*), etc. [2,6].

Typically, to design analog and RF CMOS ICs, initially, by following a design methodology that uses, for instance, the ratio of the transconductance ($gm$) by the drain current ($I_{DS}$) by the ratio of $I_{DS}$ normalized by the aspect ratio ($W/L$, where $W$ and $L$ are the width and length channel of MOSFET, respectively) methodology ($gm/I_{DS}$ as a function of $I_{DS}/(W/L)$) [5], the designers obtain several first-order equations that describe the electrical behaviors of these ICs. From these equations it is possible to obtain a first solution ($W$ and $L$ of MOSFETs and bias conditions). This first solution is used to perform the first simulation, usually by using the Simulation Program with Integrated Circuit Emphasis (SPICE). After that, the designer verifies whether all the specifications and figures of merit (output variables) are met. If not, only one of the input variables ($L$, $W$, bias conditions, etc.) must be changed by the designer and a new SPICE simulation must be performed in order for the designer to verify how the specifications behave as a function of this changed input variable. Thus, the designers can discover how the CMOS IC responds to this change in order to perform other SPICE simulations after it uses this information. Therefore, it can gradually understand more about the behavior of CMOS ICs until it is able to carry out the intended project. This process usually needs to be repeated many times until all desired design specifications and figures of merit are achieved satisfactorily. We can see that the accuracy of the solution found with this approach most of the time is not satisfactory, since the concept of human knowledge-based sizing forces the use of the first-order analytical equations. Other problems with this approach are the significant increase in DOCT, the hard work required of designers to develop the analog and RF CMOS ICs, and the difficulty in designing the analog and RF CMOS ICs with different technologies [7,8].

In order to overcome these problems, by using aggregation functions (fitness functions), evolutionary optimization heuristics algorithms (EAs) of artificial intelligence (AI) are commonly used to boost the design and optimization processes of analog and RF CMOS ICs because these methodologies do not depend on first-order analytical equations and automatically adjust the dimensions of MOSFETs in order to meet the constraints and objectives desired by the designers [8]. In addition, the evolutionary algorithms of AI can be used to trigger SPICE simulations for CMOS ICs until the desired specifications are found. There are currently three distinct categories of EAs that can be used in the optimization of analog and RF CMOS ICs: a priori, a posteriori, and progressive [9]. The a priori methodology usually define the desired specifications to be found by EAs before the optimization process. The a posteriori methodology is responsible for obtaining many different possible solutions, and after the optimization process, the designers can choose the best solution that meets the desired specifications. The progressive approach allows the designer to make changes in the design parameters during the optimization process and can be used in conjunction with both methodologies (a priori or a posteriori) [9].

During the last few years, the evolutionary-optimization heuristics algorithms of AI have been used to achieve all the desired specifications at the same time as the analog and RF CMOS IC design with robust electrical performance and a significant decrease in DOCT. For example, the authors demonstrated in [10] the advantages of using a genetic algorithm (GA) to design and optimize CMOS operational transconductance amplifiers (OTAs) with robustness. In [11], the authors designed and optimized a two-stage Miller CMOS OTA considering the 130 nm bulk CMOS IC technology node by means of an artificial intelligence heuristic approach in which they proposed a hybrid sampling method to perform the robustness analysis, with a reduced sample size and conventional random sampling. Regarding the work described in [12], the authors proposed a mono-objective metaheuristic (whale optimization algorithm) applied to the optimization of values for the aspect ratios of MOSFETs and the biasing currents of an amplifier intended for low-power low-voltage biomedical applications. The study described [13] proposed the use of multi-objective metaheuristics with an optimization algorithm inspired by butterflies to optimize analog CMOS ICs, taking into account the performance evaluation in relation to the variations of the manufacturing process, supply voltage, and temperature (robustness analysis). However, those robustness analyses are not in the loop of the design and

optimization processes of analog CMOS ICs, which can result in a considerable increase in DOCT.

Professional (commercial) electronic design automation (EDA) tools usually complete the design and optimization of analog and RF CMOS ICs in two parts. The first part is responsible for obtaining one or more nominal solutions for analog and RF CMOS ICs. The second one uses this nominal solution to perform the robustness analyses, which usually increases DOCT [14,15]. In order to reduce DOCT, another alternative EDA computational tool was created, named iMTGSPICE [16]. This computational tool is written in visual C++ language and manages SPICE simulations. This tool uses a SPICE simulator freely available on the internet, called SPICE OPUS [17]. In addition, iMTGSPICE is not associated with, or linked to, any professional EDA IC de-sign platforms available on the current market, i.e., when the designer uses iMTGSPICE, there is no need to use other professional tools to design and optimize analog and radiofrequency CMOS ICs. However, iMTGSPICE can be considered in any current design flow for analog and radiofrequency CMOS ICs. In this case, designers can validate the solutions found by iMTGSPICE with the professional EDA IC design tools used in their traditional design flow.

iMTGSPICE incorporates several AI optimization algorithms, such as a genetic algorithm (GA) [18], the shuffled frog leaping algorithm (SFLA) [19], simulated annealing (SL) [20], and the imperialist competitive algorithm (ICA) [21] to help the designers find robust possible solutions for analog and RF CMOS ICs by using corner and Monte Carlo analyses. These robustness analyses are carefully executed in the optimization loop of the design and optimization processes of analog and RF CMOS ICs without reducing the sample space of the possible solutions and without degrading the reliability and accuracy of the robustness of these CMOS ICs [18].

We observed that currently in the scientific literature, the promising evolutionary algorithm for optimization inspired by imperialistic competition (ICA) still has not been used to design and optimize analog and RF CMOS ICs with robustness analyses in the optimization loop; however, it has been used very successfully to solve many optimization problems in other engineering areas [22–24]. Therefore, the motivation of this study is to propose a customized imperialist competitive algorithm (ICA) to design and optimize robust Miller CMOS OTAs with two different bulk CMOS IC manufacturing processes from the Taiwan Semiconductor Company (TSMC) (180 nm and 65 nm nodes). The electrical performances of these two Miller CMOS OTAs were compared with other two Miller CMOS OTAs, also implemented using the same CMOS ICs technological nodes from TSMC, but designed and optimized using the customized genetic algorithm (GA) [18]. In addition, another objective of this work is to analyze the DOCT of the Miller CMOS OTAs that were designed and optimized by both ICA and GA.

## 2. iMTGSPICE Methodology with Customized ICA

A novel robustness-determined optimization process executed in an electronic design automation (EDA) tool to optimize analog and radiofrequency CMOS ICs, named iMT-GSPICE, is presented in detail in this section. It was designed in visual C++ language and combined with the SPICE simulator [17], and includes pioneering fitness functions based on the Gaussian function [16]. iMTGSPICE is capable of enhancing the effectiveness of the optimization and tolerance in relation to variations in the environmental conditions and manufacturing process of analog CMOS ICs designs. This is because it uses an evolutionary ICA algorithm [21] with nominal and robustness analyses (corner and Monte Carlo) integrated in the same optimization process. It is important to highlight that the nominal analyses are performed first, then the corner analyses, and finally the Monte Carlo analyses are carried out only for possible solutions that meet all desired specifications found by the corner analyses.

The flowchart of the customized ICA concerning the SPICE simulations that consider the robustness analyses (corner and Monte Carlo) in the loop of the optimization process is illustrated in Figure 1.

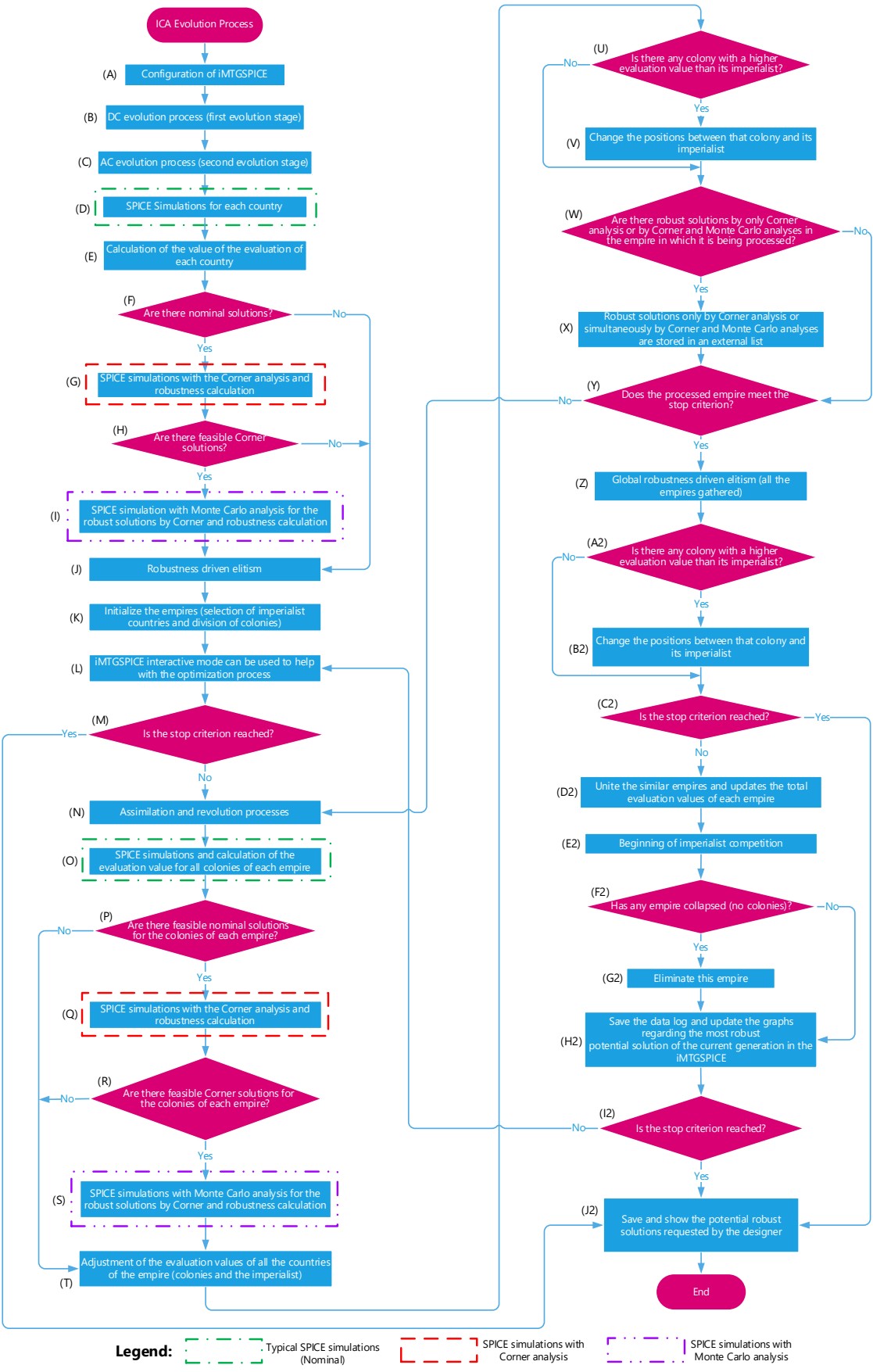

**Figure 1.** The flowchart of the customized evolutionary ICA algorithm.

Based on Figure 1, regarding Block A, iMTGSPICE must be configured. The designer must specify the following settings in iMTGSPICE: (1) the text file containing the description circuit (netlist file) of the analog CMOS IC to be optimized (simulation file with the description of the analog CMOS IC, MOSFETs dimensions, the drain current ($I_{POL}$) and the common-mode input voltage ($V_{POL}$) of the differential pair, and the technological parameters of the CMOS IC manufacturing process, which are the input design variables), (2) the desired specifications with their respective tolerance ranges (open-loop voltage gain ($A_{V0}$), unit voltage-gain frequency ($f_T$), phase margin (PM), direct current (DC), output voltage ($V_{OUT}$), power consumption ($P_{TOT}$), slew rate (SR), and total gate area of MOSFETs ($A_G$), which are the output variables), (3) customized ICA parameters ($N_{CA}$ and $N_{MCA}$, which correspond to the number of robust solutions to be found in the optimization process through SPICE simulations considering the corner and Monte Carlo analyses, respectively, which are new ICA parameters implemented in this new iMTGSPICE version; $N_{Iter}$ is the maximum number of iterations to be evolved (stop criterion if $N_{CA}$ and $N_{MCA}$ is not found); $N_{Pop}$ represents the population size (number of initial solutions or countries); $N_{Imp}$ represents the number of initial imperialists; $P_{Rev}$ and $P_{IC}$ are the revolution and imperialist competition rates, respectively; $\beta$ is the assimilation coefficient; $\zeta$ is a positive number less than unity and represents the percentage contribution of the colonies' average evaluation value to the empire's evaluation value; $U_{TH}$ is the threshold of union between two empires; $We_i$ is the weight of each FoM of the fitness function, considering that $i$ is the index each FoM; and $\sigma$ is the standard deviation of the Gaussian evaluation functions [14]), And (4) to configure the minimum and maximum values of the design variables, such as MOSFETs' channel length and width, and the $I_{POL}$ and $V_{POL}$ of the differential pair of the Miller CMOS OTA.

iMTGSPICE performs a two-stage evolution process [25]. The first stage evolves the bias conditions of MOSFETs to ensure that all of them operate in the desired operation region, e.g., saturation region, and meet the specifications of the DC bias conditions of the analog CMOS IC, such as $V_{OUT}$, $P_{TOT}$, and $A_G$. This stage is called the DC evolution process and is shown in Block B (Figure 1). In the end of the DC evolution process, $N_{MCA}$ robust possible solutions are found via corner and Monte Carlo analyses that meet the desired DC specifications of the analog CMOS IC. Then, the second stage of the evolution process of iMGTSPICE begins, called the AC evolution process (Block C in Figure 1). Firstly, a set of $N_{Pop}$ random possible solutions (countries) is generated. Secondly, the possible solutions found in the first stage, which are robust per the corner and Monte Carlo analyses, are defined by $N_{MCA}$ and must replace possible solutions created randomly in this second stage of the evolution process at randomly drawn positions.

Next, the evolutionary system of iMTGSPICE performs the typical SPICE simulations for each country of the current iteration (Block D of Figure 1). The results of these SPICE simulations are the figures of merit (FoMs) of the analog CMOS IC (nominal wanted specifications: $A_{V0}$, $f_T$, PM, $V_{OUT}$, $P_{TOT}$, SR, and $A_G$), considering the average parameters of the CMOS IC manufacturing process, usual operating conditions, and a temperature equal to 25 °C (298 K) in these SPICE simulations. The design variables (geometric dimensions and bias conditions of each MOSFET) and FoMs of these possible solutions are stored in the computer memory, so this SPICE simulation is not executed again to reduce the time of ICA evolution process, i.e., in this procedure iMTGSPICE uses a strategy referred to as reuse of the possible solutions previously found.

The value of the fitness function ($Eval(FoM_i)$) of each FoM found by the optimization process is designed according to (1), as shown in Block E of Figure 1. The range of values measured for Equation (1) is from 0, which represents a FoM out of specification, to 100, when a FoM meets the desired specification by the designer. Furthermore, the evaluation value of each FoM depends on the profile that the designer chooses for the fitness function, which can be of minimization (e.g., reducing the value of $P_{TOT}$), center value (e.g., the ideal value of PM is the middle value of a range of values specified by the designer),

or maximization (e.g., achieving a value of $A_{V0}$ greater than the desired specification) profile [16].

$$Eval(FoM_i) = 100 \, exp\left(-\frac{\varepsilon_{Typ(i)}^2}{2 \, \sigma^2}\right), \qquad (1)$$

where $i$ represents the index of FoM; $\sigma$ is the standard deviation of the Gaussian evaluations function, which is indicated by the designer; and $\varepsilon_{Typ(i)}$ represents the relative error of FoM found through SPICE simulation concerning the desired specification, which can be calculated by Equation (2):

$$\varepsilon_{Typ(i)} = \frac{Perf_{Typ(i)} - Spec_{(i)}}{Spec_{(i)}}, \qquad (2)$$

where $Perf_{Typ(i)}$ is the nominal performance trends of an FoM found through the SPICE simulation and $Spec_{(i)}$ is the wanted value of the specification.

The calculation of the fitness function of a possible solution ($Eval_{Sol}$), considering all the FOMs found, is performed by the weighted sum of the values of FoMs and their corresponding weights ($We_i$, which are defined by the designer), according to Equation (3) [16].

$$Eval_{Sol} = \sum_{i=1}^{N_{FoM}} Eval(FoM_i) \, We_i, \qquad (3)$$

where $N_{FoM}$ is the total number of evaluated FoMs.

Block F of Figure 1 verifies whether there are possible solutions that meet all desired specifications by using typical SPICE simulations. If there are no feasible nominal solutions, the optimization process jumps to Block J. Otherwise, if feasible solutions exist, iMTGSPICE triggers SPICE simulations considering the corner analyses of these possible solutions, as shown in Block G, which are best evaluated by the fitness function given by Equation (3), whose maximum number of selected solutions is limited by $N_{CA}$. To assess whether a solution is robust, the evolutionary process verifies whether the maximum and minimum values of each FoM obtained through the corner analysis are between the maximum and minimum specification values. Posteriorly, a robustness value is calculated for each solution evaluated by corner analysis. Then, iMTGSPICE calculates the relative errors of each FoM of each possible solution, which is obtained through the corner analysis. Depending on the type of fitness function (central value, minimization, or maximization [16], the relative errors are obtained as follows: (I) central value: two relative errors are calculated, because this profile is defined by minimum, nominal, and maximum values; (II) maximization: only one error is calculated, because this profile type is described only by the desired minimum specification; (III) minimization: one relative error is calculated, because this profile is defined by the desired maximum specification. One of the errors computed for a central-value specification is expressed by the minimum value of FoM subtracted from the chosen specification, and the other one is expressed by the maximum value of FoM subtracted from the desired specification. Likewise, the error estimated for a maximization or minimization specification is outlined by the maximum or minimum FoM value subtracted from the preferred specification, correspondingly. In addition, the relative error contemplated for an FoM ($\varepsilon_{WC(i)}$) in the central-value profile of the fitness function is the one that gives the highest value between the two relative errors found. Afterwards, for the non-robust possible solutions, iMTGSPICE determines the value of a new fitness function, named $Eval(FoM)'$, which reflects these relative errors. This approach aims to penalize FoMs for the possible solutions found with the highest relative errors, calculated by Equation (4). Consequently, it is possible to reduce the probability of the least robust possible solutions

being part of an empire as the imperialist (best solution) in the next iteration of the ICA evolution process.

$$Eval(FoM_i)' = 100 \, exp\left(-\frac{\varepsilon_{WC(i)}^2}{2\,\sigma^2}\right), \tag{4}$$

In the sequence of the ICA evolution process, the values of the fitness functions are recalculated using Equation (3) to be studied in the robustness-driven elitism process, considering the possible solutions that are not robust (EvalSol). For robust possible solutions, the relative errors of each FoM are calculated following the same method defined for the non-robust solutions; however, their fitness-evaluation values are not penalized according to Equation (4). Moreover, iMTGSPICE calculates the robustness value of the robust possible solutions ($D_{PS}$). This value is calculated by means of the average value of the relative deviations of the robust possible solutions, considering the minimum and maximum values of each FoM, in relation to the desired specifications of the designer. It is important to emphasize that the signal negative (–) or positive (+) of the relative deviation of each FoM is also considered. When the obtained FoM improves the desired specification, we have a signal that is (+); otherwise, when the obtained FoM is worse than the desired specification, we have a signal that is (–). Therefore, the greater the deviation ($D_{PS}$), the better its robustness value. The objective of this approach is to perform the ranking of the best robust possible solutions found. If the total number of robust possible solutions is not achieved, which is defined by the $N_{CA}$ parameter, iMTGSPICE saves the robust possible solutions found so far in an external list (Best_Sol_C) to be used to compose the next iteration through the assimilation, revolution, and imperialist competition processes. If more than $N_{CA}$ robust possible solutions are obtained, they are ordered from the highest to the lowest robustness value ($D_{PS}$) and the least robust possible solutions are excluded from the list of robust solutions by the corner analysis (Best_Sol_C).

The minimum and maximum values of FoMs and the design variables (channel length, channel width, and bias conditions of MOSFETs) of these possible solutions are also saved in computer's memory. This strategy avoids performing repeated corner analyses in future iterations if any possible solution from the previous iteration is used to compose the next iteration. This procedure (reuse strategy) is often used because the ICA evolution process always ponders the best-possible evaluated solution(s) (whereas iMTGSPICE has not found one robust possible solution yet) or the most robust possible solution(s) obtained in the preceding iteration(s) assessed. Besides, in this study all countries of the population that are possible solutions that were not estimated earlier are stored in computer's memory along with their equivalent performances (values of *L* and *W*, bias conditions, figures of merit, and robustness found by SPICE simulations). The number of stored solutions in the computer memory can be limited by the designer.

Block H of Figure 1 verifies whether there are feasible solutions resulting from the corner analyses, i.e., the maximum and minimum values of FoM obtained through the SPICE simulations by means of the corner analyses are between the minimum and maximum specification values. If there are no feasible solutions from the corner analyses, the optimization process jumps to Block J. Otherwise, if there are, iMTGSPICE triggers the SPICE simulations considering the Monte Carlo analyses of these possible solutions, as shown in Block I, which are the best-evaluated robust solutions from the list from the corner analysis (Best_Sol_C), where the maximum number of solutions to be selected to perform the Monte Carlo analyses is limited by $N_{MCA}$. Similarly, in the procedure performed for the corner analyses in Block G, a value for the robustness is calculated for each solution evaluated by Monte Carlo analysis. Then, iMTGSPICE determines the relative errors of each FoM for each possible solution considering their profiles (central value, maximization, minimization), which is obtained through the Monte Carlo analysis. Likewise, to the procedure performed in the corner analysis, for the Monte Carlo simulations the relative error considered for an FoM is given by $\varepsilon_{WC(i)}$. Subsequently, for the possible solutions that are not robust, considering the Monte Carlo analysis, iMTGSPICE also calculates the new fitness-function value for each FoM (*Eval(FoM)'*). This procedure is performed to

penalize FoMs of the possible solutions found with the highest relative errors, according to Equation (4), to reduce the probability of less-robust possible solutions being part of an empire as an imperialist (best solution) in the next iteration of the ICA evolution process. Subsequently, iMTGSPICE recalculates the values of the fitness functions of the non-robust possible solutions ($Eval_{Sol}$) through Equation (3) and, consequently, these solutions are examined by the robustness-driven elitism process. For robust possible solutions, the relative errors in each FoM are calculated following the same method described for the solutions that are not robust; however, their fitness-evaluation values are not penalized according to Equation (4). Besides, the robustness value of the robust possible solutions ($D_{PS}$) is calculated, and the higher the deviation, the better the value of its robustness. This approach ranks the best-possible robust solutions found. Considering that iMTGSPICE does not reach the total number of robust possible solutions, which is defined by $N_{MCA}$, iMTGSPICE saves the robust possible solutions found so far in an external list (Best_Sol_MC) to be used to compose the next iteration through the assimilation, revolution, and imperialist competition processes. If more than $N_{MCA}$ robust possible solutions are obtained, they are ordered from the highest to the lowest robustness value ($D_{PS}$) and the least robust possible solutions are removed from the list of robust solutions by the Monte Carlo analysis (Best_Sol_MC). The minimum and maximum values of FoMs and the design variables (W, L, and bias conditions of the MOSFETs) of these possible solutions are also saved in computer's memory. Consequently, it is possible to avoid performing repeated Monte Carlo analyses in the future in case any possible solution from the previous iteration is used to compose the next iteration.

In Block J of Figure 1, iMTGSPICE performs the robustness-driven elitism process. If the corner and Monte Carlo analyses are not selected by the designer, the elitism process is not carried out and the standard procedure of ICA is adopted. If only robust solutions via the corner analyses are available, iMTGSPICE selects the possible solutions that present the highest robustness (highest $D_{PS}$) to continue the ICA evolution process, in which the maximum number of the robust solutions to be found is limited by the $N_{CA}$ parameter. If the robust solutions obtained by the Monte Carlo analyses are available, the possible solutions that present the highest robustness values are designated to continue the ICA evolution process, which are limited by the $N_{MCA}$ parameter. The possible solutions that present the highest robustness values from the Monte Carlo analyses are selected to continue the evolution process, where the maximum number of robust solutions to be found is limited by $N_{MCA}$ because $N_{MCA}$ solutions must be robust considering both corner and Monte Carlo analyses, and before performing the Monte Carlo analysis, all design specifications must be met by the corner analysis. However, if a possible solution is robust only by the corner analysis and it is not robust by the Monte Carlo analysis, the elitism process considers only the robust possible solutions by corner analysis during the optimization process until it meets the robustness regarding both methods (corner and Monte Carlo). Afterwards, the population is reordered from the highest to smallest fitness-function values ($Eval_{Sol}$).

Afterwards, the most robust possible solutions found by corner analysis or Monte Carlo analysis must replace the possible solutions of the current iteration that present the smallest values of fitness functions. The most robust possible solutions found through the Monte Carlo analysis (or solutions found by corner analysis, if Monte Carlo solutions are not available) must replace the possible solutions with smallest values of fitness functions of the current iteration. Then, the reordering of the current population is performed, in which the value of the fitness function of each individual is changed regarding its robustness. Therefore, the evaluation values are assigned according to the robustness rating. The most robust solutions receive the highest evaluation values, and then the best nominal solutions receive the remaining evaluation values from the population. In contrast, if there are no robust solutions previously stored in the computer's memory, the solution that presented the highest value of the fitness function in the population before the robustness analyses is identified. If the fitness-function value *EvalSol*, obtained by Equation (3), was downgraded through the fitness function driven by robustness in Equation (4), the original

fitness function value is restored. This procedure avoids the stagnation of the evolution process while there are no robust solutions.

In Block K of Figure 1, the empires are initialized. Firstly, $N_{Imp}$ imperialist countries (best solutions) are selected. Then, the remaining countries (colonies) are divided among the imperialists (colonies are randomly drawn). The number of colonies $N_{Col}$ assigned for each imperialist is proportional to its power (evaluation value) [21].

Posteriorly, the designer can pause the optimization process to change the design parameters and parameters of the evolutionary algorithm and optimization process by using iMTGSPICE interactive mode, as described in Block L of Figure 1.

In Block M, the evolutionary algorithm checks whether the solutions already evaluated meet the stop criterion, which in this case is to obtain a certain number ($N_{MCA}$) of solutions that meet all specifications with the corner and Monte Carlo analyses. If the initial population evaluated meets the stop criterion, iMTGSPICE saves and shows the robust possible solutions requested by the designer, i.e., the optimization process jumps to Block J2. Otherwise, each empire is evolved until the stopping criterion is met, which starts in Block N of Figure 1. In this block, firstly the assimilation process is performed for each colony of an empire. This ICA operator is responsible for moving each colony towards the imperialist of the empire. To carry out this procedure, the distance ($d$) of each colony in relation to the imperialist is calculated, and considering $\beta$ equal to 2, a distance between 0 and $\beta \cdot d$ is drawn for each colony so that each colony can reach both sides of the imperialist [21]. Then, some colonies of each empire may go through a process called revolution. This operator is similar to the mutation operator of GA and is responsible for exploring the search space for possible solutions with better efficiency (search for the best global solutions). Firstly, the number of colonies in each empire that undergo a revolution process is calculated by the product of the revolution rate and the total number of colonies in the empire ($N_{Col}$). Next, the revolting colonies (solutions) are randomly generated and randomly replace some colonies in the empire [26].

Subsequently, the nominal SPICE simulations are performed for all colonies of each empire and the evaluation value of each colony is calculated, as described by Block O of Figure 1. In the next step, Block P is responsible for verifying whether there are feasible nominal solutions for the colonies of each empire. In an affirmative case, in Block Q the SPICE simulations are performed with the corner analysis and then the robustness of these solutions is calculated. The maximum number of solutions to be evaluated is limited by the $N_{CA}$ parameter. This block penalizes the evaluation value of non-robust corner solutions as a function of the maximum error of each specification in relation to the desired value, as described in Block G. The robust solutions found in the current iteration are saved in an internal list (local Best_Sol_C) of the empire (local list), in which the number of robust solutions desired by the designer ($N_{CA}$) is respected. If the number of corner solutions found is greater than $N_{CA}$, the excess less-robust solutions are removed. Otherwise, if there are no feasible nominal solutions for the colonies of each empire, the optimization process jumps to Block T.

Block R in Figure 1 verifies whether there are feasible solutions obtained by the corner analyses for the colonies of each empire. If none have been found, the optimization process jumps to Block T, as Figure 1 illustrates. Otherwise, SPICE simulations are performed with the Monte Carlo analysis and then the robustness of these solutions is calculated, as shown in Block S. The maximum number of solutions to be evaluated is limited by $N_{MCA}$. This block penalizes the evaluation value of non-robust Monte Carlo solutions as a function of the maximum error of each specification in relation to the desired value, as described in Block G. The robust solutions found in the current iteration are saved in an internal list (local Best_Sol_MC) of the empire (local list), in which the number of robust solutions desired by the designer ($N_{MCA}$) is respected. If the number of Monte Carlo solutions found is greater than $N_{MCA}$, the excess less-robust solutions are removed.

Considering Block T, the evaluation values of the countries of the empire are adjusted. This stage assigns evaluation values according to the robustness ranking. The most robust

solutions receive the highest evaluation values, and then the best nominal solutions receive the remaining evaluation values from the population. At the end of this process, if there is no robust solution, it does not penalize the best nominal solution found if it does not reach robustness.

Following the flowchart in Figure 1, Block U verifies whether there is any colony with a higher evaluation value than its imperialist. In an affirmative case, the positions between this colony and its imperialist are changed, as indicated in Block V of Figure 1. Otherwise, the position of the imperialist is not changed and the optimization process jumps to Block W.

Block W checks whether there are robust solutions by only corner analysis or by corner and Monte Carlo analyses in the empire that is being processed. If this condition is true the optimization process goes to Block X of the flowchart in Figure 1, and if there are robust solutions by the corner analysis in the considered empire (local Best_Sol_C), such solutions are copied to global list Best_Sol_C (global list Best_Sol_C is responsible for storing all corner solutions of the population, which preserves only the number of robust solutions desired by the user ($N_{CA}$)). If the number of corner solutions found is greater than $N_{CA}$, the excess less-robust solutions are removed. Posteriorly, the robust solutions by the Monte Carlo analysis in the empire (local Best_Sol_MC) that are being analyzed are copied to global Best_Sol_MC (global list Best_Sol_MC is responsible for storing all Monte Carlo solutions of the population, which keeps only the number of robust solutions desired by the user ($N_{MCA}$)). If the number of Monte Carlo solutions found is greater than $N_{MCA}$, the additional less-robust solutions are excluded. When the condition of Block W is not true, the optimization process jumps to Block Y, as shown in Figure 1.

Subsequently, the optimization process verifies whether the empire that is being analyzed meets the stop criterion in Block Y, as Figure 1 illustrates. The stop criterion occurs when $N_{CA}$ and $N_{MCA}$ are found; if they are not found, the stop criterion occurs when the maximum number of iterations ($N_{Iter}$) is reached. If the stop criterion is not reached, the optimization process returns to Block N and continues the process of evolution of the remaining empires of the current iteration, as described in Figure 1. However, in an affirmative case, the global robustness-driven elitism [18] (all empires gathered) is performed in Block Z. In this block, the optimization process assigns the evaluation values according to the classification of robustness. The most robust solutions receive the highest evaluation values, and then the best nominal solutions receive the remaining evaluation values from the population. Finally, the worst solutions in the population, which encompasses all empires, are preferentially replaced by existing robust solutions by Monte Carlo analysis, and if they do not exist, the existence of robust solutions is verified by the corner analysis that is used in this replacement process. However, if there are no robust solutions by the Monte Carlo and corner analyses, the replacement process is not carried out.

Continuing the description of the flowchart in Figure 1, Blocks A2 and B2 present the same functions as Blocks U and V, respectively, i.e., if there is a colony with a higher evaluation value than the imperialist of the empire, their positions are changed.

Block C2 verifies whether the stop criterion was reached. If it was, the optimization process jumps to Block J2, as illustrated in Figure 1. If this is condition false, similar empires are united [26] and the total evaluation values of each empire are updated, as shown in Block D2 in Figure 1. The union procedure between two similar empires is carried out as follows. Initially, the size of the search space is calculated by the vector formed by the difference between the maximum value and the minimum value of each input variable (transistor dimensions, bias voltages, bias currents, etc.). Then, the modulus (size) of this vector $D_{SS}$ is calculated. The value of the distance, considered the union threshold between two empires $D_{TH}$, is given by the product of the size of the search space ($|D_{SS}|$) by the parameter that represents the union threshold ($U_{TH}$)—that is, $D_{TH} = |D_{SS}| \cdot U_{TH}$. Then, the distance between two consecutive imperialists, which are part of two empires, is calculated through the vector formed by the difference between the values of each input variable

obtained by the imperialists of these empires ($D_{Imp}$). Then, the magnitude of the vector of the distances between the imperialists $|D_{Imp}|$ is calculated. If the distance between two imperialists is less than or equal to $D_{TH}$, the two empires are joined. In this case, the empire that has the strongest imperialist (highest evaluation value) receives as colonies the imperialist and the colonies of the empire that has the weakest imperialist (lowest evaluation value).

The total evaluation value of each empire is given by the imperialist's evaluation value added to a percentage of the average evaluation value of their colonies, as shown in Equation (5) [21]:

$$Eval_{TOT(i)} = Eval\left(Imp_{(i)}\right) + \zeta \; mean\left[Eval\left(Col_{(i)}\right)\right], \tag{5}$$

where $Eval_{TOT(i)}$ is the total evaluation value of the empire of index *i*, $Eval(Imp_{(i)})$ is the evaluation function value of the imperialist of index *I*, and $mean[Eval(Col_{(i)})]$ is the average evaluation value of the colonies of the empire of index *i*.

Considering Block E2, the imperialist competition [21] is started, as Figure 1 illustrates. In this competition, one of the empires that takes possession of other colonies is selected according to its probabilities (the more powerful empire is more likely to be chosen). The probability of each empire being chosen in this competition is obtained by dividing the evaluation value of each empire (calculated by Equation (5)) by the total sum of the evaluation value of all empires. Afterwards, a draw is carried out among all the empires, where the one with the highest evaluation value is the most likely to be chosen [21]. Similar to the crossover rate of the genetic algorithm, the imperialist competition occurs with a certain incidence rate ($P_{Comp}$). A rate of 11% was adopted in the experiments carried out in this work. Then, the selected empire conquers one of the weakest empire's colonies (randomly drawn). Next, Block F2 verifies whether the weakest empire has collapsed, i.e., whether there is one colony or none in this empire. In such a case, the imperialist of the weakest empire is attached to the colonies of the selected empire and the weakest empire is eliminated, as illustrated in Block G2. If the weaker empire has not collapsed, the flowchart moves to Block H2, as illustrated in Figure 1.

Posteriorly in Block H2 of Figure 1, iMTGSPICE saves the log and mapping files and update graphs regarding the most robust possible solution found in the current iteration. The input variables, such as the geometric dimensions and bias conditions of MOSFETs; design specifications such as voltage gain, power consumption, etc. of the integrated circuit; and the value of the fitness function ($Eval_{Sol}$) are plotted in graphics in real time in the Performances tab of iMTGSPICE, allowing the designer to observe how the optimization process is being carried out to meet the desired specifications.

Next, Block I2 in Figure 1 has the function of stopping the ICA optimization process by verifying whether the $N_{CA}$ and $N_{MCA}$ values have been found or whether the $N_{Iter}$ value has been reached. When these two conditions are not achieved, a new iteration is performed to try to improve the colonies of the current empires. Therefore, the optimization process goes back to Block L, where the designer has the opportunity to pause the optimization process to help the artificial intelligence to improve the next iteration of ICA algorithm even more.

Finally, Block J2 in Figure 1 is responsible for providing the most robust possible solutions found through ICA optimization process at the end of the iMTGSPICE search process.

## 3. Miller CMOS OTA Architecture and Desired Design Specifications

The Miller CMOS OTA is known as a basic analog building block, which is widely used to implement amplifiers. It features all the performance characteristics of an OTA, but it has a high open-loop voltage gain for low frequencies in a single amplifier stage. However, this type of amplifier is characterized by having two amplification stages. The first stage is a differential amplifier and the second is an inverting amplifier. A compensation capacitor is placed between the two stages in order to maintain the stability of the integrated circuit

at high frequencies [27]. Furthermore, the Miller CMOS OTA has a low output impedance at its operating frequency [28].

Figure 2 illustrates the schematic of the Miller CMOS OTA architecture that will be used in this work.

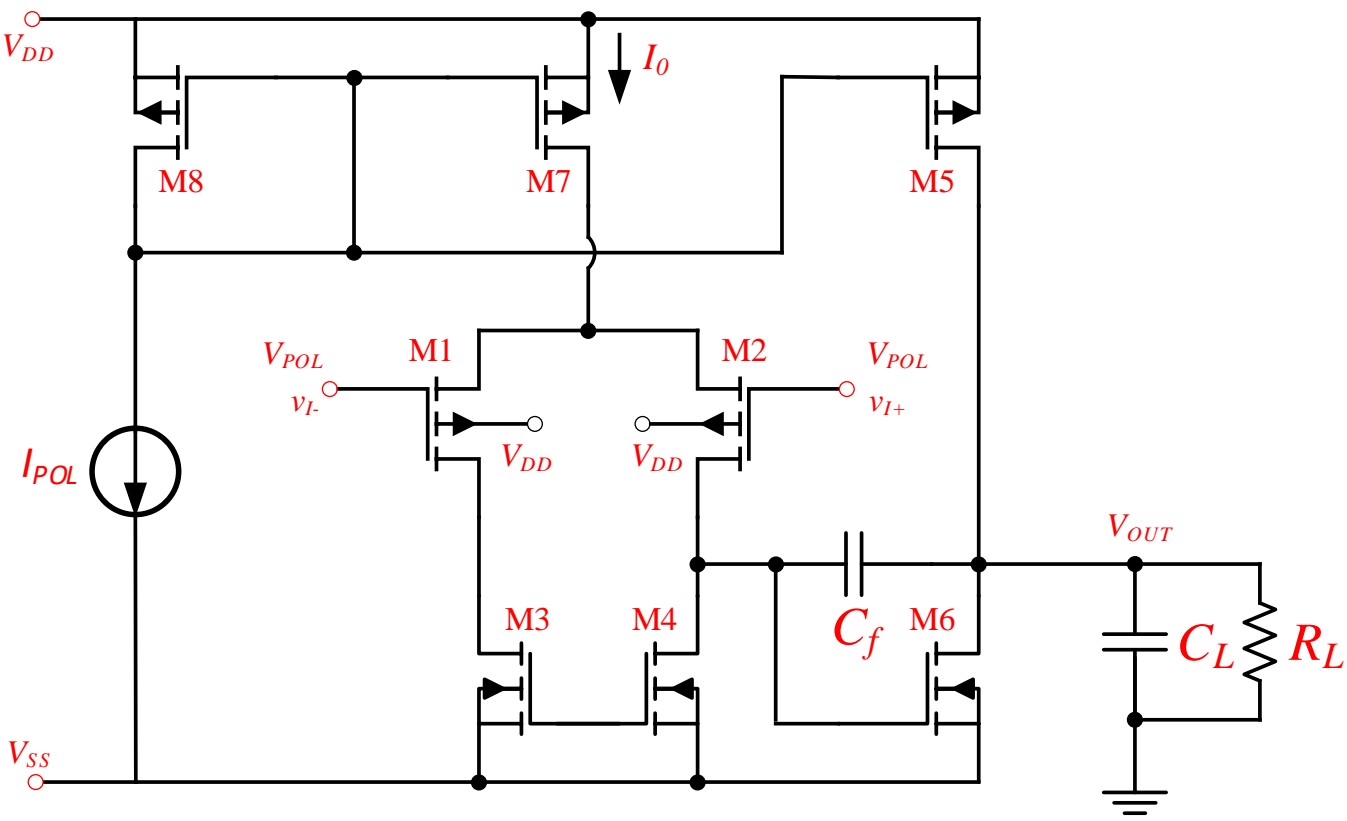

**Figure 2.** The Miller CMOS OTA architecture designed and optimized in this study.

In Figure 2, $V_{DD}$ and $V_{SS}$ are positive and negative symmetrical supply voltages. $I_{POL}$ is the Miller CMOS OTA bias current, which is implemented by an external current source; $V_{POL}$ is the common mode bias voltage that must be applied to the differential inputs; $v_{I+}$ and $v_{I-}$ are the differential inputs, where $v_{I+}$ is the non-inverting input terminal and $v_{I-}$ is the inverting input terminal; $V_{OUT}$ is the direct current (DC) output voltage; $C_f$ is the compensating capacitor; and $C_L$ and $R_L$ are the capacitive and resistive loads, respectively. The body terminals of n-channel MOSFETs (nMOSFETs) are connected to $V_{SS}$ and the body terminals of p-channel MOSFETs (pMOSFETs) are connected to $V_{DD}$. The first stage of Miller CMOS OTA consists of the matched differential pair M1/M2 (pMOSFETs) and their corresponding active loads M3/M4 (nMOSFETs) operating in the current mirror configuration, which are also matched. The output stage (second stage) is composed of M5 (pMOSFET) and M6 (nMOSFET). The M7/M8 and M5/M8 pairs (pMOSFETs) operate as current mirrors to bias the first and second stages, respectively. $C_f$ is connected between the first and second stages and is used for stability considerations (phase-margin adjustment).

Table 1 presents the minimum, nominal, and maximum values adopted for the desired design specifications of each Miller CMOS OTA (implemented with the bulk CMOS ICs manufacturing processes of 180 nm and 65 nm from TSMC) that were set in iMTGSPICE, which proposes evaluating the main design targets of the Miller CMOS OTA simultaneously: open-loop voltage gain ($A_{V0}$), unit voltage-gain frequency ($f_T$), phase margin (PM), voltage-gain margin (GM), power-supply rejection ratio (PSRR), slew rate measured in the rising edge of the signal ($SR_R$), slew rate measured in the falling edge of the signal ($SR_F$), $V_{OUT}$, $P_{TOT}$, and gate area of MOSFETs ($A_G$).

**Table 1.** The desired design specifications of the Miller CMOS OTAs for the bulk CMOS ICs manufacturing processes of 180 nm and 65 nm technology nodes from TSMC.

| Design Specifications | Bulk CMOS ICs Manufacturing Processes from TSMC | | | | | |
| --- | --- | --- | --- | --- | --- | --- |
| | Technology of 180 nm | | | Technology of 65 nm | | |
| | Minimum | Nominal | Maximum | Minimum | Nominal | Maximum |
| $A_{V0}$ | 70 dB | - | - | 70 dB | - | - |
| $f_T$ | 2 MHz | - | - | 2 MHz | - | - |
| PM | 45° | 60° | 90° | 45° | 60° | 90° |
| GM | 20 dB | - | - | 20 dB | - | - |
| PSRR | 60 dB | - | - | 60 dB | - | - |
| $SR_R$ | 2 V/µs | - | - | 2 V/µs | - | - |
| $SR_F$ | 2 V/µs | - | - | 2 V/µs | - | - |
| $V_{OUT}$ | 2.25 V | 2.5 V | 2.75 V | 1.485 V | 1.65 V | 1.815 V |
| $P_{TOT}$ | - | - | 1.2 mW | - | | 0.6 mW |
| $A_G$ | - | - | 5000 µm² | - | | 5000 µm² |

The $V_{DD}$ values were predefined in both bulk CMOS ICs technologies (equal to 5 V for technology of 180 nm and 3.3 V for technology of 65 nm). The $V_{POL}$ was set to $V_{DD}/2$ in order to ensure that all MOSFETs of the Miller CMOS OTA operate in the saturation region. The $C_L$, $R_L$ and $V_{SS}$ values are equal to 50 pF, 1 MΩ, and 0 V, respectively, in both CMOS ICs technologies of TSMC. Furthermore, the design variables were set as follows: (1) the geometric variables of nMOSFETs and pMOSFETs (channel length ($L$) and channel width ($W$)) were set to present dimensions that are proportional to the grid of the design rules of the bulk CMOS ICs manufacturing processes of 180 nm (0.60 µm $\leq L \leq$ 10 µm e 0.22 µm $\leq W \leq$ 20 µm) and 65 nm (0.50 µm $\leq L \leq$ 10 µm e 0.40 µm $\leq W \leq$ 20 µm) from TSMC used in each Miller CMOS OTA design, aiming to facilitate their layouts' implementations; (2) the value of Miller CMOS OTA bias current ($I_{POL}$) was established to be within the range of 1 µA to 30 µA; and (3) the value of compensating capacitor ($C_f$) was set to be included in the range of 1 pF $\leq C_f \leq$ 6 pF.

## 4. Results and Discussion

The iMTGSPICE simulations were run in a personal computer equipped with an Intel® Core™ i7-10700 with 2.90 GHz clock and 16 GB RAM. The operating system used was the Windows 11 Home Single Language (64 bits).

The searching process (evolution process) of the robust possible solutions was carried out using iMTGSPICE through the two customized optimization algorithms of AI, i.e., ICA and GA. The evolution process was carried out in two stages. The first stage is called DC evolution, in which the DC analysis is responsible for evolving the bias conditions of MOSFETs to ensure that all operate in the saturation region and meet the specifications of the DC bias conditions of the analog CMOS IC, such as $V_{OUT}$ and $P_{TOT}$. In this phase, the gate areas of MOSFETs can be considered in the evolving process. The DC evolution allows iMTGSPICE to speed up the search process of the second stage. Posteriorly, the second stage performs the small-signal analysis, and this phase is called alternating current (AC) evolution. The AC evolution is defined as the searching process of solutions in which all the desired design targets are evaluated ($A_{V0}$, $f_T$, PM, GM, $P_{TOT}$, etc.). Besides, during these two stages of the evolution process (DC and AC), the robustness analyses (corner and Monte Carlo) are performed automatically for each possible solution [18].

Table 2 presents the ICA and GA optimization-parameter values to perform the design of the Miller CMOS OTA in both bulk CMOS ICs manufacturing processes from TSMC

(180 nm and 65 nm CMOS IC technology nodes), where the GA optimization parameters are crossover rate ($P_C$) and mutation rate ($P_M$). In both optimization processes we used the maximum number of iterations, equal to 5000, with the number of robust possible solutions equal to 1 for corner and Monte Carlo analyses ($N_{CA}$ and $N_{MCA}$ are equal to 1), and the number-of-runs parameter was set to be equal to 1. Consequently, at the end of DC and AC evolution processes, iMTGSPICE provides one robust possible solution for the corner and Monte Carlo analyses.

**Table 2.** GA and ICA optimization parameters used to design the Miller CMOS OTAs.

| GA Optimization Parameters (DC and AC Analysis) | | ICA Optimization Parameters (DC and AC Analysis) | |
|---|---|---|---|
| Population size ($N_{Pop}$) | 40 | Number of countries ($N_{Pop}$) | 40 |
| Crossover rate ($P_C$) | 70% | Number of initial imperialists ($N_{Imp}$) | 3 |
| Mutation rate ($P_M$) | 3% | Revolution rate ($P_{Rev}$) | 10% |
| | | Imperialist competition rate ($P_{IC}$) | 11% |
| | | Assimilation coefficient ($\beta$) | 2 |
| | | Zeta ($\zeta$) | 2% |

In order to perform a fair comparison between the design and optimization processes of the Miller CMOS OTAs by using ICA and GA customized evolutionary algorithms integrated in iMTGSPICE, we performed 20 optimization processes for both evolutionary algorithms (ICA and GA), and each experiment started with the same seed in DC evolution, which was randomly generated at the beginning of each simulation, i.e., the same initial population was used by each experiment in both evolutionary algorithms [18]. At the end of each simulation, iMTGSPICE presented one robust solution for the Miller CMOS OTA, which was robust after both corner and Monte Carlo analysis. Thus, iMTGSPICE obtained 20 robust possible solutions to design and optimize each Miller CMOS OTA, considering both bulk CMOS IC manufacturing processes from TSMC (180 nm and 65 nm nodes).

After all the optimization processes, iMTGSPICE was capable of performing a more detailed robustness analyses and obtaining reports to analyze detailed information regarding the results of FoMs obtained by nominal SPICE simulations and robustness analyses (corner or Monte Carlo, and the user can generate a separated report for each of them) regarding the 20 robust possible solutions considered in this study foe each Miller CMOS OTA. The report files generated by the iMTGSPICE rank all obtained possible solutions according to their robustness, where the first classified is considered the most robust solution (in this study, the 20 obtained possible solutions for each Miller CMOS OTA were classified). Thus, Table 3 presents the most robust solutions for each Miller CMOS OTA designed and optimized by iMTGSPICE using ICA and GA customized evolutionary algorithms, taking into account the bulk CMOS IC manufacturing processes of 180 nm (Table 3a) and 65 nm (Table 3b) from TSMC.

Based on Table 3, it is possible to observe that iMTGSPICE was capable of designing and optimizing robust Miller CMOS OTAs in both customized evolutionary algorithms (ICA and GA) considered because all desired design specifications were reached (according to Table 1) after the robustness analyses (corner and Monte Carlo analyses). The ICA and GA showed the capability to enhance all desired design specifications with a maximum difference of 18% and 25% between the two customized evolutionary algorithms when the Miller CMOS OTA was implemented with the 180 nm and 65 nm CMOS ICs manufacturing processes, respectively, and this percentage difference was mainly caused by the fact that the slew rate at the falling edge of the output signal ($SR_F$) and phase margin ($PM$) was smaller and higher, respectively, when the GA customized evolutionary algorithm was used in relation to that obtained by the ICA customized evolutionary algorithm.

**Table 3.** The most robust solutions for each Miller CMOS OTA designed and optimized by iMT-GSPICE with ICA and GA customized evolutionary algorithms, which were implemented using the 180 nm (a) and 65 nm (b) CMOS IC manufacturing processes.

(a)

| Design Specifications | Miller CMOS OTA Implemented with the 180 nm CMOS IC Manufacturing Processes of TSMC | | | | | | |
|---|---|---|---|---|---|---|---|
| | | | | Customized Evolutionary Algorithms | | | |
| | Minimum | Nominal | Maximum | ICA | | GA | |
| | | | | Corner Analysis | Monte Carlo Analysis | Corner Analysis | Monte Carlo Analysis |
| $A_{V0}$ (dB) | 70 | - | - | 77.98 | 80.81 | 77.92 | 81.13 |
| $f_T$ (MHz) | 2 | - | - | 2.37 | 2.87 | 2.17 | 2.60 |
| PM (°) | 45 | 60 | 90 | 56.64 | 58.15 | 54.84 | 57.42 |
| GM (dB) | 20 | - | - | 23.48 | 23.64 | 25.87 | 26.25 |
| PSRR (dB) | 60 | - | - | 89.86 | 77.96 | 78.97 | 80.67 |
| $SR_R$ (V/μs) | 2μs | - | - | 2.38 | 2.36 | 2.12 | 2.07 |
| $SR_F$ (V/μs) | 2μs | - | - | 6.09 | 6.01 | 4.98 | 4.95 |
| $V_{OUT}$ (V) | 2.25 | 2.5 | 2.75 | 2.5 | | | |
| $P_{TOT}$ (mW) | - | - | 1.2 | 0.65 | 0.78 | 0.58 | 0.74 |
| $A_G$ (μm²) | - | - | 5000 | 1026 | | 2165 | |
| Average DOCT (min) | - | - | - | 16.32 | | 15.37 | |

(b)

| Design Specifications | Miller CMOS OTA Implemented with the 65 nm CMOS IC Manufacturing Processes of TSMC | | | | | | |
|---|---|---|---|---|---|---|---|
| | | | | Customized Evolutionary Algorithms | | | |
| | Minimum | Nominal | Maximum | ICA | | GA | |
| | | | | Corner Analysis | Monte Carlo Analysis | Corner Analysis | Monte Carlo Analysis |
| $A_{V0}$ (dB) | 70 | - | - | 79.49 | 82.87 | 77.47 | 80.49 |
| $f_T$ (MHz) | 2 | - | - | 2.42 | 2.94 | 2.01 | 2.43 |
| PM (°) | 45 | 60 | 90 | 47.88 | 47.81 | 58.93 | 59.83 |
| GM (dB) | 20 | - | - | 22.97 | 22.98 | 27.60 | 27.90 |
| PSRR (dB) | 60 | - | - | 82.79 | 80.07 | 81.94 | 83.92 |
| $SR_R$ (V/μs) | 2 | - | - | 2.14 | 2.18 | 2.15 | 2.26 |
| $SR_F$ (V/μs) | 2 | - | - | 4.12 | 4.07 | 3.48 | 3.69 |
| $V_{OUT}$ (V) | 1.485 | 1.65 | 1.815 | 1.65 | | | |
| $P_{TOT}$ (mW) | - | | 0.6 | 0.41 | 0.47 | 0.35 | 0.45 |
| $A_G$ (μm²) | - | | 5000 | 3306 | | 1546 | |
| Average DOCT (min) | - | - | - | 18.29 | | 33.45 | |

The values of $A_G$ in the four most robust solutions (Table 3) were smaller than the desired specifications mainly due to using ICA in the bulk CMOS IC manufacturing processes of 180 nm and using GA in the bulk CMOS ICs manufacturing processes of 65 nm. Besides, the values of *W* and *L* of all MOSFTEs of each Miller CMOS OTA, optimized by iMTGSPICE, were within the desired specifications range.

However, when the ICA customized evolutionary algorithm was used to design and optimize a Miller CMOS OTA with robustness, it was capable of reducing DOCT by up to 83% in relation to that implemented with the GA customized evolutionary algorithm, keeping practically the same electrical performances and robustness of these two amplifiers, especially when the Miller CMOS OTA was implemented with the most sophisticated bulk CMOS IC manufacturing process, which in this case was the 65 nm one, as can be observed in Table 3b. Therefore, it is possible to conclude that the ICA customized evolutionary algorithm has a greater ability to reduce the DOCT of a Miller CMOS OTA with robustness in relation to the use of the GA customized evolutionary algorithm.

Table 4 presents the values of the geometric dimensions of all transistors (length and channel width) and their multiplicities, the values of the internal compensation capacitor ($C_f$), and the bias current ($I_{POL}$) for the most robust solutions obtained by the ICA and GA customized evolutionary algorithms for each Miller CMOS OTA and in two different bulk CMOS ICs manufacturing processes (180 nm and 65 nm technology nodes), which are presented in Table 3.

**Table 4.** MOSTFETs' design variables regarding the most robust solutions for each Miller CMOS OTA designed and optimized by iMTGSPICE with ICA (a) and GA (b) customized evolutionary algorithms, which were implemented in the 180 nm and 65 nm CMOS ICs manufacturing processes.

(a)

| Design Variables (Input Variables) | MOSFET Design Variables of the Miller CMOS OTA Designed and Optimized by iMTGSPICE with the ICA Customized Evolutionary Algorithm | | | | | | | | | | | |
|---|---|---|---|---|---|---|---|---|---|---|---|---|
| | 180 nm Bulk CMOS IC Technology Node | | | | | | 65 nm Bulk CMOS IC Technology Node | | | | | |
| $0.22\ \mu m \leq W \leq 20\ \mu m$ | M1/M2 | M3/M4 | M5 | M6 | M7 | M8 | M1/M2 | M3/M4 | M5 | M6 | M7 | M8 |
| | 2.92 | 18.07 | 16.70 | | 2.99 | 7.86 | 8.19 | 15.08 | 14.23 | 7.28 | 19.96 | 10.27 |
| $0.60\ \mu m \leq L \leq 10\ \mu m$ | 4.35 | 2.21 | 0.65 | | 1.23 | | 8.84 | 0.52 | 1.10 | 1.43 | 8.58 | |
| Multiplicity (number of MOSFETs in parallel) | 10 | 3 | 19 | 18 | 9 | 10 | 10 | 3 | 8 | 20 | 4 | 9 |
| $1\ \mu A \leq I_{POL} \leq 30\ \mu A$ | 20.50 | | | | | | 11.00 | | | | | |
| $1\ pF \leq C_f \leq 6\ pF$ | 1.30 | | | | | | 2.20 | | | | | |

(b)

| Design Variables (Input Variables) | MOSFET Design Variables of the Miller CMOS OTA Designed and Optimized by iMTGSPICE with GA Customized Evolutionary Algorithm | | | | | | | | | | | |
|---|---|---|---|---|---|---|---|---|---|---|---|---|
| | 180 nm Bulk CMOS ICs Technology Node | | | | | | 65 nm Bulk CMOS ICs Technology Node | | | | | |
| $0.40\ \mu m \leq W \leq 20\ \mu m$ | M1/M2 | M3/M4 | M5 | M6 | M7 | M8 | M1/M2 | M3/M4 | M5 | M6 | M7 | M8 |
| | 9.69 | 2.54 | 17.75 | 17.16 | 8.06 | 11.51 | 16.19 | 6.44 | 15.93 | 8.45 | 13.07 | 16.25 |
| $0.50\ \mu m \leq L \leq 10\ \mu m$ | 9.36 | 1.17 | 0.65 | 0.85 | 2.28 | | 7.74 | 0.65 | | 0.78 | 3.9 | |
| Multiplicity (number of MOSFETs in parallel) | 8 | 8 | 11 | 16 | 4 | 9 | 2 | 5 | 18 | 20 | 6 | 6 |
| $1\ \mu A \leq I_{POL} \leq 30\ \mu A$ | 25.50 | | | | | | 5.00 | | | | | |
| $1\ pF \leq C_f \leq 6\ pF$ | 1.55 | | | | | | 1.30 | | | | | |

Analyzing Table 4, it is possible to observe that the four most robust solutions for each Miller CMOS OTA designed and optimized by iMTGSPICE met the design rules of both manufacturing processes, taking into account the values of *W* and *L*. Furthermore, the values of $C_f$ and $I_{POL}$ of these solutions were in conformity with the established ranges for the design and optimization of Miller CMOS OTAs.

For the purpose of illustration, the robustness features of the FoMs of the Miller CMOS OTAs were plotted using box plots for the ICA and GA customized evolutionary algorithms implemented in iMTGSPICE and two different bulk CMOS IC manufacturing processes from TSMC (180 nm and 65 nm technology nodes). In addition, to generate these box plots, we also performed SPICE simulations with the Monte Carlo analysis with 5000 samples for each of the four most robust solutions that iMTGSPICE obtained (two robust possible solutions for each evolutionary algorithm), considering 100 global variations and 50 local variations. Therefore, Figure 3 illustrates the relative deviations in percentage of each FoM of the Miller CMOS OTAs in relation to their desired specifications (according to Table 1), which were obtained from SPICE simulations, regarding the AC analysis with the Monte Carlo analyses of the most robust solutions of each Miller CMOS OTA using the ICA and GA customized evolutionary algorithms.

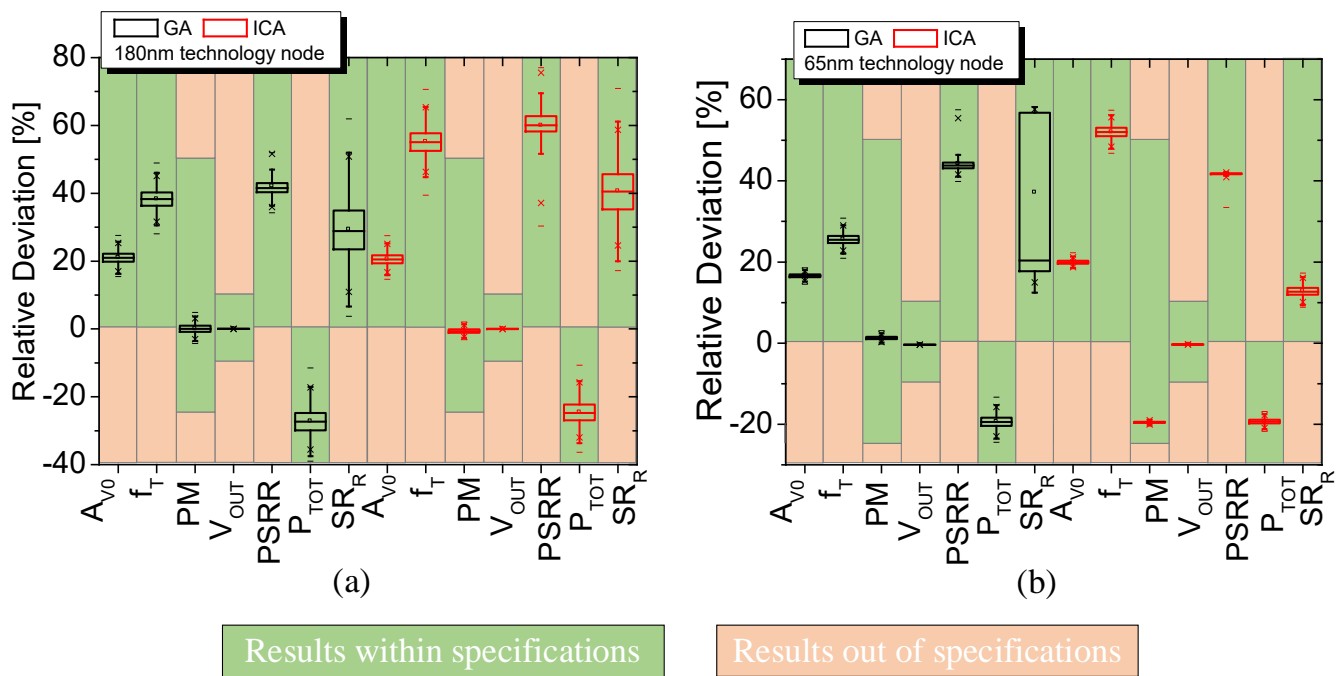

(a)                                        (b)

Results within specifications                    Results out of specifications

**Figure 3.** The relative deviations in percentage of each FoM in relation to the desired specifications (box plots) of the Miller CMOS OTAs implemented in the 180 nm (**a**) and 65 nm (**b**) bulk CMOS ICs manufacturing processes, which were obtained from the SPICE simulations with Monte Carlo analyses of the four most robust possible solutions using the ICA and GA customized evolutionary algorithms.

Analyzing Figure 3, it is possible to observe that the yields of all designs of the Miller CMOS OTAs were equal to 100% because all ICA and GA optimization processes ensured that the minimum and maximum values of all FoMs obtained through the SPICE simulations with the Monte Carlo analyses were within the tolerance ranges of the desired specifications, as described in Table 1. Besides, Figure 3a illustrates that the relative deviations of $A_{V0}$, $V_{OUT}$, *PM*, and $P_{TOT}$ of the Miller CMOS OTAs, optimized by ICA and GA, were similar in both customized evolutionary algorithms; however, ICA was capable of boosting more values of $f_T$, *PSRR*, and $SR_R$ in relation to those values found by GA. When the ICA and GA customized evolutionary algorithms are used to perform the design and optimization processes of the Miller CMOS OTAs on the 65 nm bulk CMOS ICs technology node, ICA enhanced the typical values of $A_{V0}$ and $f_T$ in relation to those presented by GA (approximately 17% and 43% higher, respectively), as can be seen in Figure 3b. On the other hand, GA was able to achieve higher values of *PM*, *PSRR*, and $SR_R$ than those found by ICA, although extremely smaller variations in these figures of merit were found using the ICA, as illustrated in Figure 3b. In addition, the relative deviations of $V_{OUT}$ and $P_{TOT}$

were similar regardless of the customized evolutionary algorithm used, but with smaller variations in $P_{TOT}$ when ICA was used, as illustrated in Figure 3b.

In addition, the robustness value of each robust possible solution ($D_{PS}$) was calculated. It was obtained through the average value of the relative deviations of the robust possible solution, taking into account the minimum and maximum values of each FoM in relation to the desired specifications. Considering the main FoMs presented in the box plots for the Miller CMOS OTAs implemented with the 180 nm bulk CMOS ICs technology node (Figure 3a), ICA obtained a greater robustness ($D_{PS}$ = 15.6%) than that presented by GA ($D_{PS}$ = 10.7%). However, the Miller CMOS OTAs implemented with the 65 nm bulk CMOS IC technology node (Figure 3b) presented similar robustness when they were optimized by the GA or ICA customized evolutionary algorithm ($D_{PS}$ = 14.8%).

Therefore, we concluded that by using the ICA and GA customized evolutionary algorithms to perform the design and optimization processes of the Miller CMOS OTAs, it is possible to achieve all FoMs within the tolerance ranges of the desired specifications, as described in Table 1, which were defined by the designer in the initial configuration of iMTGSPICE. However, when the designer chooses to use the ICA customized evolutionary algorithm to design and optimize a Miller CMOS OTA in iMTGSPICE, it is possible to reduce the design and optimization-cycle times by up to 83% in relation to that implemented with the GA customized evolutionary algorithm. There are two possibilities to explain why ICA is more capable of reducing the cycle time of the optimization process of analog CMOS IC designs in comparison to that achieved by the customized GA: the first possibility is related to the capacity of ICA be more efficient in performing a global exploratory search, due to the revolution operator that is responsible for exploring a greater diversity of solutions in the search space, taking into account those that are not close to the imperialist countries, especially considering evolution rate values above 10%. Consequently, the assimilation operator of ICA can be considered more efficient because it explores places in the search space that would not be possible to verify previously. Therefore, the colonies tend to approach imperialist country in other regions that were not previously explored in the global search space and, consequently, the search for the best solution becomes faster, reducing the DOCT. On the other hand, the mutation operator of GA, which is analogous to revolution operator of ICA, with values above 10% (experimentally verified), is able to perform a global search process for solutions that tend to become increasingly random, because the mutation process, which it is responsible for changing the chromosome genes of the possible solutions, is a random process and, consequently, the efficiency of the algorithm changes depending on the mutation rate, with a tendency to decrease algorithm efficiency with a high mutation rate. However, considering low values for the mutation rate, for instance below 10%, the global search process benefits from a broader search, but this benefit occurs slowly because it requires numerous iterations (increasing the number of generations to find the best solutions). The second possibility is related to the elitist process, i.e., whereas GA always maintains a single best solution to participate of the next generation, ICA always starts the optimization process with the number of best solutions found equal to $N_{imp}$ (in this work $N_{imp}$ was equal to three), which are maintained until the collapse of the less powerful empires occurs, therefore with only one main empire remaining (the most powerful one), which corresponds to the best solution found, which is the imperialist of the empire.

Besides, the ICA customized evolutionary algorithm is capable of boosting the robustness (approximately 5%) of a Miller CMOS OTA in relation to the robustness presented by using the GA customized evolutionary algorithm in the optimization process, taking into account the 180 nm bulk CMOS ICs technology node, due to the higher $D_{PS}$ value obtained by ICA in relation to that presented by GA—that is, ICA tends to boost FoMs more than GA.

## 5. Conclusions

This paper proposes, for the first time, the use of a custom imperialist competitive algorithm (ICA) in order to reduce the design and optimization-cycle times of the analog CMOS ICs, with robustness analyses (corner and Monte Carlo) in the optimization loop, by using the iMTGSPICE tool. The ICA customized evolutionary algorithm was used to carry out the design and optimization processes of a Miller CMOS OTA implemented in a 180 nm bulk CMOS ICs technology node and another in a 65 nm bulk CMOS IC technology node. In addition, we performed the same design and optimization processes of the Miller CMOS OTAs by using a GA customized evolutionary algorithm in order to compare which of the two customized evolutionary algorithms presented the lowest design and optimization-cycle times. The results show that the ICA and GA customized evolutionary algorithms were capable of performing the design and optimization processes of the Miller CMOS OTAs, and it was possible to achieve all FoMs within the tolerance ranges of the desired specifications of the designer in a few minutes (on average 17 and 24 min for the ICA and GA customized evolutionary algorithms, respectively). However, the ICA customized evolutionary algorithm was capable of reducing the design and optimization-cycle times by up to 83% in relation to those implemented with the GA customized evolutionary algorithm, keeping practically the same electrical performances for the Miller CMOS OTAs implemented with the most sophisticated bulk CMOS IC manufacturing process used in this work, which was the 65 nm one. In addition, the ICA customized evolutionary algorithm was capable of improving by approximately 5% the robustness of the Miller CMOS OTA in relation to the robustness presented by the GA customized evolutionary algorithm in the optimization process for the 180 nm bulk CMOS IC manufacturing process. Therefore, iMTGSPICE can be considered as an alternative and differentiated computational tool based on the integration of artificial and human intelligences to design and optimize analog CMOS ICs to build robust blocks with a significant time reduction and consequently remarkable design-costs reduction to launch new electronic products.

**Author Contributions:** Conceptualization, E.H.S.G., S.P.G. and R.A.d.L.M.; Data Curation, E.H.S.G. and R.A.d.L.M.; Formal Analysis, E.H.S.G., S.P.G. and R.A.d.L.M.; Funding acquisition, S.P.G. and R.A.d.L.M.; Investigation, E.H.S.G. and R.A.d.L.M.; Methodology, E.H.S.G. and R.A.d.L.M.; Project Administration, E.H.S.G., S.P.G. and R.A.d.L.M.; Supervision, S.P.G. and R.A.d.L.M.; Visualization, E.H.S.G.; Writing—original Draft, E.H.S.G.; Resources, R.A.d.L.M.; Software, R.A.d.L.M.; Validation, S.P.G. and R.A.d.L.M.; Writing—review and editing, S.P.G. and R.A.d.L.M. All authors have read and agreed to the published version of the manuscript.

**Funding:** This research was funded by the São Paulo Research Foundation (FAPESP), grant number 2020/09375-0, and the National Council for Scientific and Technological Development (CNPq), grant number 307804/2019-4.

**Conflicts of Interest:** The authors declare no conflict of interest.

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
