# Peer review of "Customized Imperialist Competitive Algorithm Methodology to Optimize Robust Miller CMOS OTAs"

_electronics, doi:10.3390/electronics11233923_

Round 1
Reviewer 1 Report
easy to find in internet resources but some aditional info on iMTGSPICE would be appreciated. Info on iMTGSPICE availability for professional EDA IC design platforms .
Author Response
First of all, the Authors would really like to thank the Reviewer and Editor for helping us to improve the quality of this paper. Thank you so much for that.
In this file, the Authors address all the questions, comments, recommendations and suggestions of the Reviewer and Editor so important to improve the quality of this manuscript. Thank you so much.
The Authors are using the blue color to indicate the answers for the questions/comments/recommendations/suggestions of the Reviewer 1. Besides, this color will be also used in the file named “Changes_in_Colors.pdf” to indicate the changes added in the final version of this paper, taking into account the comments/recommendations/suggestions of each Reviewer.

Reviewer 2 Report
In the paper by Egon Henrique Salerno Galembeck, Salvador Pinillos Gimenez and Rodrigo Alves de Lima Moreto entitled CUSTOMIZED IMPERIALIST COMPETITIVE ALGORITHM METHODOLOGY TO OPTIMIZE ROBUST MILLER CMOS OTAs, the authors describe the use of a custom Imperialist Competitive Algorithm (ICA) in order to reduce the design and optimization cycle times of the analog CMOS ICs. In this study they implement some Miller CMOS Operational Transconductance Amplifiers (OTAs) by using the computational tool named iMTGSPICE, considering two different Bulk CMOS ICs manufacturing processes from Taiwan Semiconductor Company (TSMC) (180nm and 65nm nodes) and two optimization evolutionary methodologies of Artificial Intelligence, i.e., ICA and Genetic Algorithm (GA). The main result obtained by the work have showed that, by using ICA customized evolutionary algorithm to perform the design and optimization processes of Miller CMOS OTAs, it is possible to reduce the design and optimization cycle times by up to 83% in relation to those implemented with the GA customized evolutionary algorithm, achieving practically the same electrical performances.
The flowchart of the ICA customized evolutionary algorithm is illustrated in Figure 1.
The Miller CMOS OTA is known as a basic analog-building block, which is widely used to implement amplifiers; actually it is an unbuffered two stage operational amplifier. It features all the performance characteristics of an OTA, but it has a high open-loop voltage gain for low frequencies in a single amplifier stage. This type of amplifier is characterized by having two amplification stages. The first stage is a differential amplifier and the second is an inverting amplifier. A compensation capacitor is placed between the two stages to maintain the stability of the integrated circuit at high frequencies. In zddition, the Miller CMOS OTA has a low output impedance at its operating frequency. Figure 2 illustrates the schematic of the Miller CMOS OTA architecture that will be used in this work.
Table 1. reflects the desired design specifications of the Miller CMOS OTAs for the Bulk CMOS ICs manufacturing processes of 180nm and 65nm technologies nodes from TSMC.
Table 2. yields GA and ICA optimization parameters used to design the Miller CMOS OTAs.
Table 3. relects the most robust solutions for each Miller CMOS OTA designed and optimized by iMTG-SPICE with the customized evolutionary algorithms ICA and GA, respectively, which were implemented by using the 180 nm (a) and 65 nm (b) CMOS ICs manufacturing processes.
It is a detailed described useful work and can be considered for publication in the journal after performing the following improvements. I suggest strongly to add the typical transistor dimensions used or simulations in an additional table for both realization technologies of 180nm and 65nm. This is necessary for the convenience of the readers, although some explanations ar given in the text; without this information the work remains incomplete.
Author Response
First of all, the Authors would really like to thank the Reviewer and Editor for helping us to improve the quality of this paper. Thank you so much for that.
In this file, the Authors address all the questions, comments, recommendations and suggestions of the Reviewer and Editor so important to improve the quality of this manuscript. Thank you so much.
The Authors are using the red color to indicate the answers for the questions/comments/recommendations/suggestions of the Reviewer 2. Besides, this color will be also used in the file named “Changes_in_Colors.pdf” to indicate the changes added in the final version of this paper, taking into account the comments/recommendations/suggestions of each Reviewer.

Reviewer 3 Report
In this work a new Algorithm is proposed in order to reduce the design and optimization cycle times of the analog CMOS ICs. The work reported here seems novel and interesting.
to improve the quality of reported work some more literature review is required for same types of work.
the OTA included here with 180nm and 65nm, but now a days we talking about even a more smaller size. so that can be included
Author Response
First of all, the Authors would really like to thank the Reviewer and Editor for helping us to improve the quality of this paper. Thank you so much for that.
In this file, the Authors address all the questions, comments, recommendations and suggestions of the Reviewer and Editor so important to improve the quality of this manuscript. Thank you so much.
The Authors are using the green color to indicate the answers for the questions/comments/recommendations/suggestions of the Reviewer 3. Besides, this color will be also used in the file named “Changes_in_Colors.pdf” to indicate the changes added in the final version of this paper, taking into account the comments/recommendations/suggestions of each Reviewer.
